# Alcohol use disorders are associated with higher healthcare expenditure among older adults with suspected cognitive impairment: A registry-based cross-sectional study

Ben Kamsvaag[1,2☯*], Sverre Bergh[1,3☯], Jūratė Šaltytė Benth[1,4,5☯],
Richard C. Oude Voshaar[6☯], Kjerstin Tevik[2,3☯], Geir Selbaek[3,7,8☯],
Anne-Sofie Helvik[2,3☯]

1 Research Centre for Age-Related Functional Decline and Disease, Innlandet Hospital Trust, Ottestad, Norway, 2 Department of Public Health and Nursing, Faculty of Medicine and Health Sciences, Norwegian University of Science and Technology (NTNU), Trondheim, Norway, 3 Norwegian National Centre for Ageing and Health, Vestfold Hospital Trust, Tønsberg, Norway, 4 Institute of Clinical Medicine, Campus Ahus, University of Oslo, Oslo, Norway, 5 Health Services Research Unit, Akershus University Hospital, Lørenskog, Norway, 6 University of Groningen, University Medical Center Groningen, Groningen, The Netherlands, 7 Department of Geriatric Medicine, Oslo University Hospital, Oslo, Norway, 8 Institute of Clinical Medicine, Faculty of Medicine, University of Oslo, Oslo, Norway

☯ These authors contributed equally to this work.
* bekvam@sykehuset-innlandet.no

## Abstract

### Background

High levels of alcohol consumption and cognitive impairment both drain our healthcare budget, but it is unknown whether alcohol use disorders (AUDs) influence healthcare costs among people with suspected cognitive impairment specifically.

### Methods

This study aims to examine the association between alcohol-related ICD-10 diagnoses and healthcare costs among 2,736 Norwegians aged ≥60 years being assessed for cognitive impairment in Norwegian specialist healthcare and included in the Norwegian Registry of Persons Assessed for Cognitive Symptoms (NorCog). Linear regression analysis was applied to assess the relationship between ICD-10 alcohol-related diagnoses and the primary outcome variable: healthcare costs. Healthcare costs one year before and one year after clinical assessment were used to account for the expected shift in healthcare use after assessment.

### Results

Median costs of healthcare use were €2,226 (Q1-Q3 1,076−4,107) one year before assessment and €2,217 (Q1-Q3 1,160−4,006) after. One year prior to NorCog assessment, participants with AUDs had approximately 50% higher costs compared

**Data availability statement:** Data cannot be shared publicly due to ethical and legal restrictions related to participant confidentiality, imposed by the Norwegian National Centre for Ageing and Health, the Norwegian Institute of Public Health, and the Data Protection Office of Innlandet Hospital Trust. The dataset consists of sensitive, in-depth quantitative data regarding participants diagnosed with cognitive impairment and/or alcohol use disorders. Furthermore, participants consented to the use of their data solely within the scope of the present study, as approved by the South-Eastern Regional Committee for Medical Research Ethics (REK). Therefore, the dataset cannot be made publicly available. Researchers who meet the criteria for access to confidential data may request access to the dataset by contacting the institutional representatives at the Norwegian Prescribed Drug Registry, who were responsible for data linkage: lmr.data@fhi.no. For questions related to data privacy or ethical oversight, inquiries may be directed to Birgit Hovde at birgit.Hovde@sykehuset-innlandet.no (Data Protection Officer, Innlandet Hospital Trust).

**Funding:** This work was supported in full by the Innlandet Hospital Trust under Grant number 150919, which was provided solely to BK (https://www.sykehuset-innlandet.no/en/) The funders had no role in study design, data collection and analysis, decision to publish, or preparation of the manuscript.

**Competing interests:** The authors have declared that no competing interests exist.

to participants without an AUD (median of €3,286 and €2,190, respectively). One year after NorCog assessment, this difference was negligible. An interaction between AUD status and time was significant, implying that post-diagnostic care for cognitive impairment may simultaneously mitigate the healthcare burden associated with AUDs or its related sequelae.

## Conclusion

Our findings indicate that alcohol consumption is a potentially important and amenable determinant of healthcare use, knowledge which could be valuable in planning treatment and care. Such knowledge could also possibly curtail the higher healthcare costs among older adults with AUDs. Thus, we urge healthcare providers to routinely ask patients about their alcohol consumption.

## Introduction

Dementia affects 50 million people worldwide and 115,000 Norwegians, numbers that will rise substantially due to an aging population [1–3]. The syndrome disrupts a person's cognitive, physical, and psychological functioning, and is a major cause of mortality [4].

Dementia also imposes a substantial economic burden, with global annual expenditure estimated at approximately one trillion € [1]. In Europe, the average annual economic cost of dementia per person was estimated at €32,507 in 2018 [5]. Similarly, the Resource Use and Disease Course in Dementia (REDIC) report estimated these annual costs at €34,743 in Norway in 2015 [6]. Considering the economic burden dementia imposes on society, knowledge of the most important cost drivers is highly relevant. The REDIC report found that functional impairment, poor physical health, cognitive impairment, more symptoms of psychosis and depression, living alone, and being female were associated with higher healthcare costs among older adults with dementia [6]. The largest cost related to dementia, however, is by far the cost of living in nursing homes [7,8].

Alcohol consumption has also been devoted attention for its impact on general health. The proportion of alcohol-attributable deaths has been found to be approximately 4.7% [4]. Several systematic reviews have found that compared to abstention or light-to-moderate alcohol consumption, heavier consumption is associated with poorer health in a variety of ways, including an increased risk of hypertensive heart disease, hemorrhagic stroke, several cancers including oropharyngeal, laryngeal, esophageal, and colorectal cancer, liver cirrhosis, and all cause-mortality [9–15].

Of particular relevance for the present study, alcohol consumption is related to the risk of developing cognitive impairment and dementia. Traditionally, research has found either a U- or J-shaped relationship, in which consuming >168 grams of alcohol per week is associated with an increased risk of developing dementia compared to abstinence, and lighter drinking is associated with reduced risk

compared to abstinence [16–24]. However, more recent research has indicated that using abstinence as a reference category likely introduces confounding and biased estimates of the true effects of alcohol on brain health [25]. In general, abstainers have poorer overall health than occasional drinkers [26–30], and many older adults cease drinking due to poor health [31–37]. Furthermore, several studies using abstainers neglect to consider their prior alcohol consumption, which could mask the negative effects of previous consumption [25,26,38–40]. Because many studies have neglected to account for these considerations, the precise nature of the association between alcohol consumption and dementia risk is difficult to accurately assess in observational studies. What seems more clear, however, is that consuming large amounts of alcohol (>168 grams of alcohol per week) or being diagnosed with an alcohol use disorder (AUD) is consistently associated with an increased risk for developing cognitive impairment and dementia [16–18,20–24,41,42].

Furthermore, alcohol may disproportionately affect older adults. One reason is that older adults are more vulnerable to the negative effects of alcohol due to decreases in their water-to-fat ratio and reduced liver function [43–46]. Secondly, alcohol consumption is a well-established risk factor for falls, hip fractures [47,48], and accidents [49–52]. In addition, many older adults use medications which may lead to harmful drug-alcohol interactions [53–56]. Notably, typical medications used to treat dementia such as cholinesterase inhibitors have also been implicated in potential alcohol-drug interactions [57,58]. For these reasons, older adults above the age of 64 have been recommended to limit their consumption of alcohol to 14 grams (i.e., one US standard drink) per day [59,60], which is half the amount generally recommended for the general population [61]. As a point of reference, the World Health Organization has recently asserted that there exists no threshold for safe levels of alcohol consumption [62], and the 2023 Nordic Nutrition Recommendations recommends that the general population avoids alcohol consumption altogether [63].

Thus, there are many ways in which alcohol consumption at higher levels can negatively impact a person's health. However, the association between alcohol consumption and healthcare use is not as straightforward. Some studies find a negative association between alcohol consumption and healthcare use, even when accounting for past drinking patterns [64,65]. However, the majority of the studies we have identified seem to find higher healthcare use and higher healthcare costs among people who meet criteria for an AUD [66–68], among those reporting monthly binge drinking (i.e., ≥ 70 grams of alcohol on the same occasion) [67], or those drinking more than 154–196 grams per week [69], even among current abstainers when taking into account a history of binge drinking [64,70,71].

Thus, the secular trend of older adults drinking more frequently in recent years [72–75] is worrying. As far as we know, no studies have yet examined whether alcohol consumption or the presence of AUDs are associated with healthcare use among older adults with symptoms of cognitive impairment specifically. Thus, our aim was to study the association between AUDs and healthcare use in older Norwegians with suspected cognitive impairment, defined as being assessed at specialized memory clinics. The presence of one or more AUDs (i.e., dependence, harmful use, or somatic comorbidity associated with alcohol consumption) was selected as the primary independent variable, as its status as a risk factor for dementia is more clearly established in the research literature. Healthcare use was operationalized in terms of monetary costs. Healthcare costs were assessed one year before and one year after specialist assessment for cognitive symptoms. These separate time intervals were selected because healthcare utilization is expected to differ qualitatively between these phases. Specifically, the pre-assessment phase often reflects a period of suspected cognitive impairment and diagnostic uncertainty, wherein healthcare visits typically center around symptom evaluation [76]. In contrast, the post-assessment phase reflects a clarified diagnostic conclusion and the onset of more structured follow-up and intervention [76]. Thus, this design reflects an attempt to capture the diagnostic transition and to provide an interpretable quantification of healthcare costs before and after specialist assessment. Given the association between AUDs and poorer health, we hypothesize that being diagnosed with an AUD could be associated with higher healthcare costs among people with suspected cognitive impairment.

## Materials and methods

### Study population

Older adults (≥60 years) included in the Norwegian Registry of Persons Assessed for Cognitive Symptoms (NorCog) between 2014–2018 were selected for this retrospective, cross-sectional study. The NorCog registry includes information from a standardized assessment of dementia used at specialist outpatient clinics across all of Norway. Through interviews and medical tests with the participant and their next of kin, NorCog collects sociodemographic information as well as physical, psychiatric, and cognitive functioning [77]. More detailed information regarding the assessments used in NorCog is listed in Table 1.

Fig 1 presents a flow chart of the selection process. Out of the 3,608 eligible older adults (>60 years of age) registered in NorCog between 2014–2018, a total of 2,736 were deemed fit for inclusion in the study.

### Data collection and assessments

All assessments were derived from nationwide registry systems, including the NorCog database (see above), the Norway Control and Payment of Health Reimbursement (KUHR) database, or the Norwegian Patient Registry (NPR). Linkage of these data was performed by the Norwegian Prescribed Drug Registry. All variables were assessed for multicollinearity.

**Primary outcome variable.** Healthcare costs associated with primary care were calculated based on exact costs registered in the KUHR database. No such data was available for the costs associated with specialist healthcare, so these costs were estimated based on the healthcare activity registered in the NPR database in combination with monetary costs associated with such activity. The estimation method followed the approach used in the aforementioned REDIC report [6]. A more detailed description of these calculations can be found in Table 1.

**Primary determinant of interest.** AUD was operationalized as being registered with a medical diagnosis in the NorCog, KUHR, or NPR database related to dependence, harmful use, or somatic comorbidity associated with alcohol consumption (see Table 1), according to The International Statistical Classification of Diseases and Related Health Problems 10th Revision (ICD-10) [84].

**Covariates.** All covariates were extracted from the NorCog database. The diagnostic conclusion after assessment was registered as one of the following diagnoses: Subjective cognitive impairment, SCI; mild cognitive impairment, MCI; dementia; or "other diagnoses". The "other diagnoses" category consists of conditions that do not fulfil the diagnostic criteria of the other categories, and include conditions such as mood disorders and non-degenerative cognitive diagnoses [77]. In addition, we also included the Mini-Mental State Examination (MMSE) sum score as an assessment of cognitive status [86–89]. Neuropsychiatric symptoms were assessed with the Neuropsychiatric Inventory Questionnaire (NPI-Q) [90,91]. Finally, function of daily living was assessed with the Personal Activities of Daily Living (PADL) [93]. The following sociodemographic and clinical information was obtained: participant's sex, age, number of years of education, employment status, marital status, use of domiciliary care, number of chronic diseases, the number of prescribed medications, tobacco smoking habits, and the participant's relationship with their next of kin.

Finally, we conducted sensitivity analyses employing alcohol consumption frequency as an additional primary independent variable. Alcohol consumption was assessed on an eight-point Likert scale, based on information from the next of kin. The question was, "About how often in the last 12 months did the patient drink alcohol?", and the eight response alternatives ranged from "never" to "4-7 times a week".

### Ethical approval

Collection of NorCog data was approved by the Norwegian Data Inspectorate. All NorCog participants signed an informed consent form granting future use of their data. All methods were performed in accordance with relevant regulations or in accordance with the Declaration of Helsinki, confirmed by the South-Eastern Regional Committee for Medical Research

**Table 1. NorCog assessments included in study analyses.**

| Assessments | Description |
| --- | --- |
| Costs (in Euro) associated with healthcare use one year before and after NorCog assessment [6,78–80] | Costs associated with primary care were calculated based on exact costs registered in the Norway Control and Payment of Health Reimbursement (KUHR) database. Costs associated with specialist healthcare were estimated based on the healthcare activity registered in the Norwegian Patient Registry (NPR) in combination with monetary costs associated with such activity. |
| | For specialist healthcare costs (NPR), we followed the approach used in the REDIC report [6]. REDIC calculated average costs associated with overnight hospital stays, outpatient consultations, and emergency services in Norway. The average cost of overnight stays was €1,206 in 2011, the average cost of outpatient consultations was €115 in 2012, and the average cost of emergency services was €194 in 2013. A discount rate of 3% per annum since the relevant dates was applied to these numbers in order to provide valid current costs, as is standard practice [81–83]. We used the same discount rate for each successive year until reaching the time period in which most of our participants were assessed with NorCog (median year = 2017). Thus, the values used to estimate costs associated with overnight hospital stays, outpatient consultations, and emergency services in our study were €1,440, €133, and €218, respectively. These values were multiplied by the number of medical consultations to provide crude estimates of total costs per participant. |
| Alcohol use disorder as defined in The International Statistical Classification of Diseases and Related Health Problems 10th Revision (ICD-10) [84] | All participants were registered with medical diagnoses based on the ICD-10. The presence of an alcohol use disorder (AUD) was operationalized as a participant being registered with a medical diagnosis related to dependence, harmful use, or somatic comorbidity associated with alcohol consumption. The following ICD-10 codes were used to designate a participant with an AUD: F10 Mental and behavioral disorders due to the use of alcohol; G31.2 Degeneration of nervous system due to alcohol; G62.1 Alcoholic polyneuropathy; K70 Alcoholic liver disease; T51.0 Toxic effect of ethanol. |
| | The following ICD-codes were also considered for inclusion due to their relevance to alcohol use: I42.6, K29.2, K86.0, Z50.2, and Z72.1. However, for these codes, the fourth character of the diagnostic code had not been registered (e.g., the alcohol qualifier.1 for the diagnosis Z72.1) and it was therefore not possible to establish a link to alcohol use. Therefore, these ICD-codes were not included. |
| Alcohol consumption [85] | Single question in NorCog assessing frequency of alcohol consumption, based on question from the Third Nord-Trøndelag Health Study (HUNT3, 2006–2008). Based on information from next of kin. The question was, "About how often in the last 12 months did the patient drink alcohol?", and response alternatives ranged from: "Never," "Not at all last year", "A few times a year", "Once a month," "2-3 times a month", "Once a week", "2-3 times a week", to "4-7 times a week". |
| Mini-Mental State Examination – Norwegian Revised Version (MMSE-NR2/3) [86–89] | Screening instrument for cognitive impairment, Norwegian revisions. Twenty items, score range 8–30 for both NR2 and NR3 revisions. Traditionally, a score of 24 has been the cut-off value for indicating cognitive impairment. High scores = better cognitive functioning. In NorCog, a score of 24 is used as a cut-off for the indication of cognitive impairment. |
| The Neuropsychiatric Inventory – Questionnaire (NPI-Q) [90,91] | Structured interview for neuropsychiatric symptoms, based on information from next of kin. Twelve symptoms are assessed (yes/no) and graded on a 0–3-point scale. In the current study, the 12 symptoms were categorized into three subsyndromes based on factor analysis [92]: NPI-Depression, score min-max: 0–18 (items: depression, anxiety, disturbances in appetite, apathy, motor disturbances, and night-time disturbances); NPI-Agitation, score min-max: 0–12 (items: euphoria, disinhibition, irritability, and agitation); NPI-Psychosis, score min-max: 0–6 (items: hallucinations and delusions). High scores = higher subsyndrome severity. |
| Personal Activities of Daily Living (PADL) [93] | Structured assessment of personal activities of daily living, based on information from next of kin. The six following activities are assessed and graded on a 1–5-point scale: Toileting, eating, dressing, grooming, physical movement, and bathing. In the current study, the scales were re-coded into a dichotomous scale of "loss of function" (0) and "normal functioning" (1), based on the original 1969 scoring guidelines. Score range 0–6. High scores = better functioning. |

Ethics (REK sør-øst B: 21490) and the NorCog publication board. All data was pseudonymized, and the authors did not have access to information that could identify individual participants during or after data collection. The authors gained access to NorCog data on March 25 2020 and the complete, linked dataset (including data from NPR and KUHR) on September 20 2021.

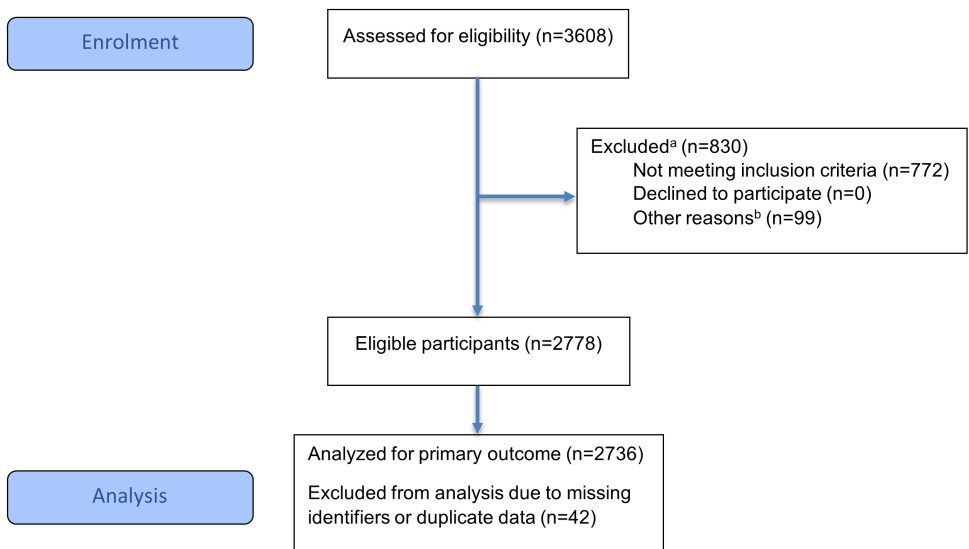

**Fig 1. CONSORT flow chart of selection process [94].** [a]Total n of excluded data in the first data pass is smaller than the sum of each source of exclusion, due to overlap. [b]99 participants were registered as deceased or were missing data regarding whether they were currently alive or deceased.

## Statistical analysis

The data was processed and analyzed with SPSS version 30, STATA version 18, and Excel. The outcome variable "healthcare costs" displayed a skewed distribution, so all values of the outcome variable were log-transformed. Cluster effect at the health institution level was assessed with the intra-class correlation coefficient (ICC). ICC was close to zero and thus there was no evidence of significant cluster effects.

A total of 2,736 participants were eligible for study inclusion (Fig 1). Due to complete data regarding ICD diagnoses, alcohol consumption, and healthcare costs being a pre-requisite for study inclusion, there were no missing values for these variables. However, due to a large number of missing values on covariates, only 1,840 could be included in the regression analyses. Comparisons between eligible (n = 2,736) and ineligible participants (n = 830) as well as between participants included and not included in regression analyses (n = 1,840 vs. n = 1,726) were performed on the following variables: Sex, age, education, MMSE-NR2/3 score, NPI-Q subsyndrome scores, and domiciliary care. Due to cluster effect in some of the variables, the comparisons were performed by generalized linear mixed model with random effects for health institution.

Missing values for the assessments PADL and NPI-Q were imputed as follows. Empirical distributions were generated for each item, and a random number drawn from this distribution was used to substitute missing values. This procedure was repeated until all imputable cases were imputed. Only cases with at least 50% non-missing values on single items of each assessment were imputed. For PADL, a total of 277 values for 177 participants were imputed. For NPI-Q, 77 values for 72 participants were imputed. In addition, missing values for marital status were imputed based on the next of kin's relationship to the participant (imputed for 8 participants).

Four linear regression models were estimated to assess the relationship between the sociodemographic and clinical variables outlined in the previous section, in particular AUDs, and the following outcome variables: (1) Primary healthcare costs one year before NorCog assessment; (2) Primary healthcare costs one year after; (3) Specialist healthcare costs one year before NorCog assessment; and (4) Specialist healthcare costs one year after. The threshold for statistical significance was set at an alpha level of.001 for all tests, due to the large sample size and number of tests performed. There was no evidence of multicollinearity for the variables listed under the Assessments section, except for the next of kin's

relationship with the participant and marital status (the latter variable was therefore excluded from the analyses). There was also a high number of missing values for the next of kin's age and sex, and these variables were also excluded. All remaining variables were adjusted for in the adjusted models. Due to a large share of missing values among the covariates, regression models with inverse probability weights were estimated.

In addition, two sensitivity analyses were conducted post hoc to examine total healthcare costs regardless of sector (primary or specialist): (1) one regression model including alcohol consumption frequency as a primary independent variable in addition to AUDs; and (2) one regression using only alcohol consumption frequency as the primary independent variable. Furthermore, to assess the robustness of the conclusions, linear mixed models with random effects for participants and fixed effects for time (one year before vs. one year after NorCog assessment), AUDs (yes vs. no) and the interaction between these two variables were estimated for each outcome. These models were also adjusted for the same variables as specified above. A significant interaction between time and AUD would imply that the association between AUD and the outcome variable one year before NorCog assessment differs from the same association one year after NorCog assessment.

## Results

The mean age was 75.9 (SD = 7.3) years, and 53.0% were female (see Table 2). In total, 52.3% were registered with a clinical diagnosis of dementia, 33.0% were registered with MCI, 4.7% with SCI, and the remaining 10.1% were categorized as "other diagnoses". The comparison of eligible (n = 2,736) and ineligible (n = 830) participants showed non-significant differences. The comparison of participants included in the regression analyses (n = 1,840) and those which were excluded due to missing covariate values (n = 1,726) showed that only age was significantly different (included participants mean 76.4 (SD = 7.1) years, excluded participants mean 75.4 (SD = 7.6) years. p < 0.001).

### Healthcare use

Median costs of total healthcare use were €2,226 (Q1-Q3 1,076−4,107) one year before NorCog and €2,217 (Q1-Q3 1,160−4,006) one year after. Mean (SD) costs were €3,090 (€2,933) one year before and €3,151 (€3,090) one year after. The median total costs of healthcare use stratified by presence of one or more AUDs (yes/no), one year before NorCog assessment and one year after, are presented in Fig 2. In descriptive terms, participants with at least one AUD had a significantly higher median total cost (€3,286) compared to those without (€2,190) the year before NorCog assessment. This difference was less pronounced for the year following directly after NorCog assessment (€2,575 and €2,209, respectively).

### Determinants of healthcare use

The results of the linear regression analysis for primary care (KUHR) and specialist care (NPR) are presented in Tables 3 and 4, respectively. Presence of an AUD was not significantly associated with primary healthcare costs (KUHR: Table 3). The interaction between time and AUD was not significant (p = .488), implying no differences in the association between AUDs and primary healthcare costs before and after cognitive assessment. In contrast, the presence of an AUD was significantly associated with higher specialist healthcare costs the year before NorCog assessment (NPR: Table 4). The year after, the associations between AUDs and healthcare costs were attenuated. Additionally, there was a significant interaction in the model for specialist healthcare, implying a significant difference in the association between AUDs and specialist healthcare costs before and after cognitive assessment (p = .009).

The results of the first sensitivity analysis, which included alcohol consumption frequency as a primary independent variable in addition to AUDs, revealed a significant association between AUDs and total healthcare costs the year prior to NorCog assessment but no significant association between alcohol consumption and total healthcare costs (S1 Table). The second sensitivity analysis, which used only alcohol consumption frequency as the primary independent variable, also found no significant associations with total healthcare costs (S2 Table).

**Table 2. Sample characteristics (n = 2,736). Numbers are n (%) if not otherwise specified.**

| Characteristics | Statistic | |
|---|---|---|
| **Participant characteristics** | | |
| Median (Q1-Q3) cost of healthcare use in Euro one year before NorCog | 2225.8 | (1076-4107) |
| Median (Q1-Q3) cost of healthcare use in Euro one year after NorCog | 2217.1 | (1160-4006) |
| Sex, female | 1451 | (53.0) |
| Mean age (SD) | 75.9 | (7.3) |
| Mean years of education (SD) | 11.0 | (3.6) |
| Employment status | | |
| Not currently working | 905 | (35.0) |
| Working 10% cent or more | 115 | (4.4) |
| Sick leave/disability benefits | 138 | (5.3) |
| Retired | 1428 | (55.2) |
| Marital status[a] | | |
| Partner (cohabited or married) | 1709 | (64.3) |
| Single | 950 | (35.7) |
| Proportion receiving domiciliary care | 937 | (34.5) |
| Mean number of chronic diseases (SD) | 2.2 | (1.7) |
| Mean number of registered medications (SD) | 4.9 | (2.9) |
| Tobacco smoking habits, as reported by next of kin | | |
| Never smoked | 1060 | (39.2) |
| Smoked previously but no longer smokes | 1261 | (46.7) |
| Currently smoking | 381 | (14.1) |
| Mean MMSE-NR2/3 score (SD) | 23.1 | (4.6) |
| Mean NPI-Q subsyndrome scores (SD) | | |
| NPI-Depression | 3.5 | (3.3) |
| NPI-Agitation | 1.3 | (1.9) |
| NPI-Psychosis | 0.6 | (1.2) |
| Mean PADL score (SD) | 4.8 | (1.5) |
| Alcohol consumption, as reported by next of kin | | |
| Never | 224 | (8.2) |
| Not at all the last year | 470 | (17.2) |
| A few times a year | 621 | (22.7) |
| Once a month | 193 | (7.1) |
| 2–3 times a month | 213 | (7.8) |
| Once a week | 276 | (10.1) |
| 2–3 times a week | 400 | (14.6) |
| 4–7 times a week | 339 | (12.4) |
| Cognitive diagnostic conclusion | | |
| Subjective cognitive impairment (SCI) | 128 | (4.7) |
| Mild cognitive impairment (MCI) | 903 | (33.0) |
| Dementia | 1430 | (52.3) |
| "Other diagnoses" | 275 | (10.1) |
| **Next of kin characteristics** | | |
| Next of kin's sex, female[b] | 894 | (66.8) |
| Next of kin's mean age (SD)[b] | 62.7 | (13.0) |

*(Continued)*

**Table 2.** (Continued)

| Characteristics | Statistic | |
|---|---|---|
| Next of kin's relationship to participant | | |
| Spouse/cohabitant | 1363 | (51.6) |
| Child/child-in-law | 1061 | (40.2) |
| Other (neighbour, friend, sibling, etc.) | 216 | (8.2) |

Percentages may not total 100 due to rounding. NorCog = Norwegian Registry of Persons Assessed for Cognitive Symptoms, SD = standard deviation, MMSE-NR2/3 = Mini-Mental State Examination – Norwegian Revised Version 2/3, NPI-Q = The Neuropsychiatric Inventory – Questionnaire, PADL = Personal Activities of Daily Living.

[a]Not included in further analyses due to multicollinearity caused by high correlation with next of kin's relationship to participant.

[b]Not included in further analyses due to high number of missing values.

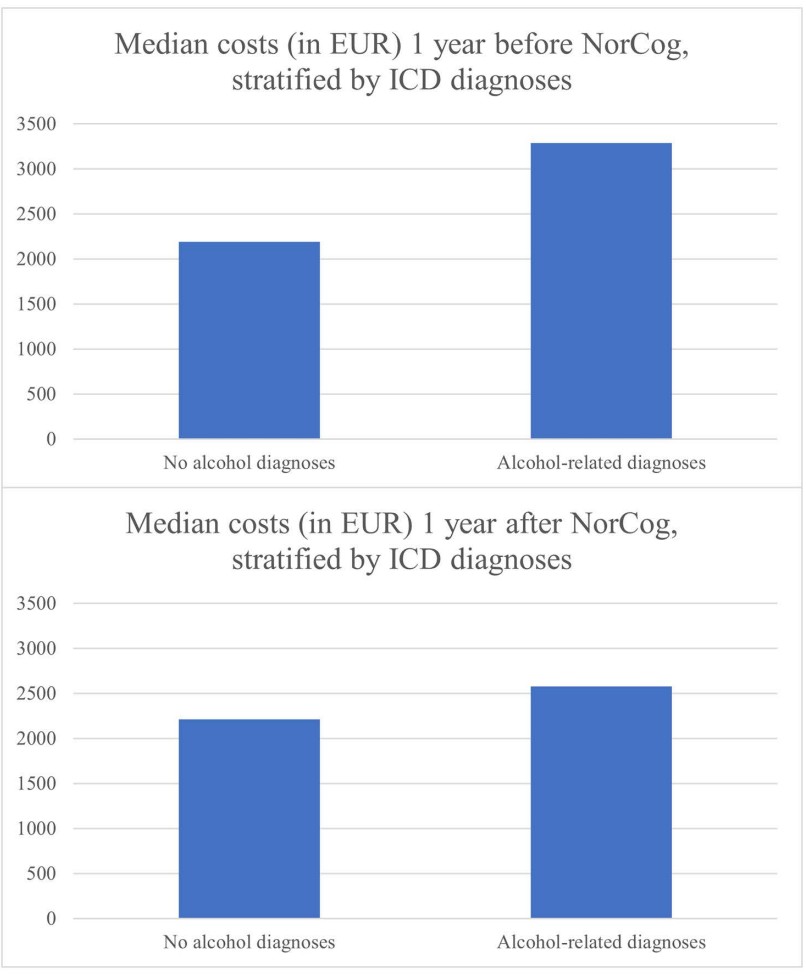

**Fig 2. Median total costs of healthcare use stratified by ICD diagnoses.** EUR = Euro.

**Table 3. Unadjusted and adjusted linear regressions of KUHR-costs one year before and after NorCog assessment.**

| Independent variables | One year before assessment | | | | One year after assessment | | | |
|---|---|---|---|---|---|---|---|---|
| | Unadjusted model | | Adjusted model | | Unadjusted model | | Adjusted model | |
| | RC (95% CI) | p | RC (95% CI) | p | RC (95% CI) | p | RC (95% CI) | p |
| *Primary independent variable* | | | | | | | | |
| Alcohol-related ICD diagnosis, Yes | .20 (.008;.39) | **.041** | .10 (−.06;.27) | .227 | .13 (−.04;.29) | .125 | .02 (−.14;.18) | .805 |
| *Covariates* | | | | | | | | |
| Age | −.002 (−.007;.003) | .412 | −.005 (−.01;.001) | .078 | −.017 (−.022; −.013) | **<.001** | −.01 (−.02; −.01) | **<.001** |
| Sex, male | .06 (−.01;.13) | .109 | .01 (−.06;.08) | .848 | .14 (.07;.21) | **<.001** | .07 (−.007;.14) | .076 |
| Education (no. of years) | .001 (−.01;.01) | .821 | .01 (−.001;.02) | .078 | .02 (.006;.03) | **.001** | .006 (−.004;.02) | .259 |
| Employment status | | | | | | | | |
| Not currently working | Reference | | Reference | | Reference | | Reference | |
| Working 10% or more | −.12 (−.31;.07) | .208 | −.06 (−.24;.12) | .490 | .13 (−.05;.32) | .155 | −.02 (−.20;.16) | .829 |
| Sick leave/disability benefits | .15 (−.02;.31) | .077 | .07 (−.09;.22) | .386 | .27 (.10;.44) | **.002** | .06 (−.12;.24) | .508 |
| Retired | .02 (−.06;.09) | .636 | .02 (−.05;.09) | .495 | −.02 (−.09;.06) | .662 | −.02 (−.09;.05) | .511 |
| Receives domiciliary care, Yes | .20 (.13;.27) | **<.001** | .12 (.04;.20) | **.003** | −.06 (−.13;.005) | .069 | −.02 (−.10;.06) | .578 |
| Number of chronic diseases | .10 (.08;.12) | **<.001** | .03 (.01;.05) | **.001** | .07 (.05;.09) | **<.001** | .04 (.01;.06) | **.001** |
| Number of medications[a] | .17 (.02)[b] | **<.001** | .15 (.02)[b] | **<.001** **<.001** | .06 (.04;.07) | **<.001** | .05 (.04;.06) | **<.001** |
| Number of medications x Number of medications[a] | −.006 (.002)[b] | **<.001** | −.006 (.001)[b] | | | | | |
| Tobacco smoking habits, as reported by next of kin | | | | | | | | |
| Never smoked | Reference | | Reference | | Reference | | Reference | |
| Smoked previously but no longer smokes | −.01 (−.08;.07) | .814 | −.06 (−.13;.01) | .104 | .006 (−.07;.08) | .866 | −.05 (−.12;.02) | .135 |
| Currently smoking | .02 (−.09;.13) | .700 | −.07 (−.18;.04) | .221 | −.07 (−.18;.03) | .179 | −.17 (−.27; −.06) | **.001** |
| MMSE-NR2/3 sum score | .02 (.01;.02) | **<.001** | .01 (.004;.02) | **.003** | .03 (.02;.04) | **<.001** | .02 (.01;.03) | **<.001** |
| NPI-Q subsyndrome scores | | | | | | | | |
| NPI-Depression score | .03 (.01;.04) | **<.001** | .02 (.01;.03) | **<.001** | .007 (−.003;.02) | .182 | .01 (−.002;.02) | .101 |
| NPI-Agitation score | .02 (.004;.04) | **.015** | −.004 (−.02;.01) | .674 | .01 (−.005;.03) | .162 | .003 (−.01;.02) | .741 |
| NPI-Psychosis score | .03 (.002;.06) | **.037** | .02 (−.01;.05) | .195 | .01 (−.02;.04) | .367 | .04 (.007;.06) | **.016** |
| PADL sum score | −.03 (−.05; −.005) | **.017** | .01 (−.02;.03) | .622 | .03 (.003;.05) | **.027** | .01 (−.01;.04) | .354 |
| Cognitive diagnostic conclusion | | | | | | | | |
| Subjective cognitive impairment (SCI) | −.05 (−.24;.13) | .581 | −.15 (−.32;.02) | .088 | .14 (−.03;.30) | .106 | −.15 (−.32;.02) | .092 |
| Mild cognitive impairment (MCI) | .19 (.11;.27) | **<.001** | .13 (.04;.21) | **.004** | .22 (.15;.30) | **<.001** | .07 (−.01;.15) | .110 |
| Dementia | Reference | | Reference | | Reference | | Reference | |
| "Other diagnoses" | .23 (.11;.34) | **<.001** | .15 (.03;.26) | **.014** | .21 (.09;.33) | **<.001** | .04 (−.08;.15) | .541 |
| Next of kin's relationship to participant | | | | | | | | |
| Spouse/cohabitant | Reference | | Reference | | Reference | | Reference | |
| Child/child-in-law | −.02 (−.09;.05) | .549 | −.06 (−.14;.02) | .138 | −.16 (−.23; −.09) | **<.001** | −.04 (−.12;.04) | .341 |
| Other (neighbour, friend, sibling etc.) | .08 (−.06;.22) | .252 | .007 (−.13;.14) | .925 | −.02 (−.16;.11) | .744 | .02 (−.11;.14) | .795 |

Outcome values were log-transformed prior to analysis. Significant p values in bold. In the adjusted models, all variables listed were adjusted for. Inverse probability weighting was employed. KUHR, the Norway Control and Payment of Health Reimbursement; NorCog, Norwegian Registry of Persons Assessed for Cognitive Symptoms; RC, Regression coefficient; CI, Confidence interval; ICD, International Classification of Diseases; MMSE-NR2/3, Mini-Mental State Examination – Norwegian Revised Version 2/3; NPI-Q, The Neuropsychiatric Inventory – Questionnaire; PADL, Personal Activities of Daily Living.

[a]"Number of medications" showed a curvilinear association with the outcome in the adjusted model of healthcare costs one year prior to NorCog.

[b]Standard error instead of CI is presented due to second-order term.

**Table 4. Unadjusted and adjusted linear regressions of NPR-costs one year before and after NorCog assessment.**

| Independent variables | One year before assessment | | | | One year after assessment | | | |
|---|---|---|---|---|---|---|---|---|
| | Unadjusted model | | Adjusted model | | Unadjusted model | | Adjusted model | |
| | RC (95% CI) | p | RC (95% CI) | p | RC (95% CI) | p | RC (95% CI) | p |
| *Primary independent variable* | | | | | | | | |
| Alcohol-related ICD diagnosis, Yes | .63 (.33;.93) | **<.001** | .51 (.23;.79) | **<.001** | .08 (−.22;.38) | .610 | −.001 (−.29;.28) | .992 |
| *Covariates* | | | | | | | | |
| Age | .007 (−.0007;.01) | .074 | −.004 (−.01;.005) | .385 | .0003 (−.007;.008) | .928 | −.006 (−.02;.004) | .221 |
| Sex, male | .19 (.08;.29) | **.001** | .13 (.03;.24) | **.014** | .22 (.11;.32) | **<.001** | .17 (.06;.28) | **.002** |
| Education (no. of years) | −.02 (−.03; −.004) | **.015** | −.005 (−.02;.01) | .508 | −.01 (−.02;.005) | .194 | −.006 (−.02;.01) | .447 |
| Employment status | | | | | | | | |
| Not currently working | Reference | | Reference | | Reference | | Reference | |
| Working 10% or more | −.41 (−.69; −.13) | **.004** | −.27 (−.54;.01) | .057 | −.29 (−.58;.007) | .056 | −.21 (−.52;.09) | .174 |
| Sick leave/disability benefits | −.0001 (−.26;.26) | .999 | −.06 (−.30;.18) | .614 | .05 (−.20;.31) | .700 | −.03 (−.28;.22) | .837 |
| Retired | −.09 (−.20;.03) | .134 | −.08 (−.18;.02) | .134 | −.15 (−.26; −.04) | **.008** | −.15 (−.26; −.05) | **.005** |
| Receives domiciliary care, Yes | .53 (.42;.64) | **<.001** | .40 (.27;.53) | **<.001** | .22 (.11;.33) | **<.001** | .05 (−.08;.19) | .426 |
| Number of chronic diseases | .17 (.15;.20) | **<.001** | .09 (.06;.12) | **<.001** | .11 (.08;.14) | **<.001** | .05 (.01;.08) | **.007** |
| Number of medications | .12 (.11;.14) | **<.001** | .07 (.05;.09) | **<.001** | .09 (.08;.11) | **<.001** | .07 (.05;.09) | **<.001** |
| Tobacco smoking habits, as reported by next of kin | | | | | | | | |
| Never smoked | Reference | | Reference | | Reference | | Reference | |
| Smoked previously but no longer smokes | .003 (−.11;.12) | .965 | −.12 (−.23; −.01) | **.026** | .07 (−.04;.18) | .208 | −.03 (−.14;.08) | .631 |
| Currently smoking | .03 (−.14;.20) | .750 | −.15 (−.32;.02) | .084 | −.09 (−.25;.07) | .280 | −.22 (−.39; −.06) | **.008** |
| MMSE-NR2/3 sum score | .01 (.003;.03) | **.012** | .006 (−.008;.02) | .386 | .02 (.005;.03) | **.004** | .02 (.005;.03) | **.006** |
| NPI-Q subsyndrome scores | | | | | | | | |
| NPI-Depression score | .03 (.01;.05) | **.001** | .01 (−.01;.03) | .292 | .03 (.01;.04) | **<.001** | .02 (−.003;.03) | .102 |
| NPI-Agitation score | .04 (.01;.07) | **.004** | .02 (−.007;.05) | .124 | .04 (.01;.07) | **.007** | .01 (−.02;.04) | .485 |
| NPI-Psychosis score[a] | .16 (.06)[b] | **.009** | .07 (.06)[b] | .226 | .04 (−.005;.08) | .085 | .01 (−.04;.06) | .632 |
| NPI-Psychosis score x Psychosis Score[a] | −.04 (.01)[b] | **.004** | −.03 (.01)[b] | **.025** | | | | |
| PADL sum score | −.13 (−.16; −.09) | **<.001** | −.07 (−.11; −.03) | **.001** | −.08 (−.12; −.05) | **<.001** | −.05 (−.10; −.01) | **.015** |
| Cognitive diagnostic conclusion | | | | | | | | |
| Subjective cognitive impairment (SCI) | .10 (−.19;.39) | .488 | .16 (−.13;.45) | .279 | −.08 (−.36;.20) | .581 | −.14 (−.43;.14) | .319 |
| Mild cognitive impairment (MCI) | .29 (.18;.41) | **<.001** | .33 (.20;.46) | **<.001** | .13 (.02;.25) | **.023** | .08 (−.04;.21) | .204 |
| Dementia | Reference | | Reference | | Reference | | Reference | |
| "Other diagnoses" | .43 (.25;.60) | **<.001** | .43 (.26;.60) | **<.001** | .21 (.02;.39) | **.030** | .14 (−.04;.33) | .132 |
| Next of kin's relationship to participant | | | | | | | | |
| Spouse/cohabitant | Reference | | Reference | | Reference | | Reference | |
| Child/child-in-law | .01 (−.10;.12) | .835 | −.13 (−.25; −.01) | **.041** | −.02 (−.13;.09) | .728 | −.01 (−.15;.12) | .834 |
| Other (neighbour, friend, sibling etc.) | .22 (.006;.43) | **.044** | .02 (−.18;.22) | .865 | .09 (−.12;.31) | .382 | .05 (−.15;.26) | .604 |

Outcome values were log-transformed prior to analysis. Significant p values in bold. In the adjusted models, all variables listed were adjusted for. Inverse probability weighting was employed. NPR, the Norwegian Patient Registry; NorCog, Norwegian Registry of Persons Assessed for Cognitive Symptoms; RC, Regression coefficient; CI, Confidence interval; ICD, International Classification of Diseases; MMSE-NR2/3, Mini-Mental State Examination – Norwegian Revised Version 2/3; NPI-Q, The Neuropsychiatric Inventory – Questionnaire; PADL, Personal Activities of Daily Living.

[a]"NPI-Psychosis score" showed a curvilinear association with the outcome in the adjusted model of healthcare costs one year prior to NorCog.

[b]Standard error instead of CI is presented due to second-order term.

## Discussion

In this study of older Norwegians being assessed for cognitive impairment, costs of healthcare use varied substantially among participants (interquartile range=€3,031). Median total costs per year were €2,226 one year before and €2,217 one year after NorCog assessment. Of primary interest, participants with at least one AUD had a significantly higher median total cost (€3,286) compared to those without (€2,190) the year before NorCog assessment, and AUDs were significantly associated with higher costs of both total and specialist healthcare use in regression models. Additionally, an interaction analysis between AUD status and time indicated that this difference significantly decreased the year after assessment, which could have important clinical implications which will be discussed in the following paragraphs. These figures amount to approximately 50% higher healthcare costs among participants with AUDs compared to those without an AUD the year prior to NorCog assessment.

While our findings corroborate earlier reports of higher healthcare costs among people who meet criteria for an AUD [66–68], we are the first to demonstrate such an effect among older adults with an AUD and simultaneous symptoms of cognitive impairment specifically. This knowledge about the role of alcohol is important, due to the many health challenges unique for older adults generally and for older adults with cognitive impairment specifically. As noted in the introduction, frequent drinking and AUD diagnoses have been implicated in a range of negative health consequences and is associated with higher healthcare use. Furthermore, many medications commonly prescribed to older adults have harmful interactions with alcohol which may lead to adverse events such as falling, hospital admission, and death [53–55,95]. Of particular interest to our sample, there are potential alcohol-medication interactions with anti-dementia drugs (e.g., cholinesterase inhibitors like donepezil) [57,58]. In light of these findings, individuals presenting with both cognitive impairment and an AUD may represent a clinically complex subgroup at elevated risk for adverse outcomes and higher healthcare costs. Alcohol consumption could potentially thereby be an important and modifiable determinant of healthcare use. This knowledge highlights the need for targeted intervention for this specific patient subgroup, and it can be valuable in planning treatment and care.

It is not clear why the difference in healthcare costs between participants with AUDs compared to those without was less pronounced the year after NorCog assessment, and we can only speculate on the reasons. One possible reason could relate to the potential impact of the NorCog assessment, which may have functioned as an intervention in and of itself. Indeed, the NorCog working group has reported that 95% of all their participants who end up with a dementia or MCI diagnosis are subsequently referred to municipal follow-up [96]. While we do not have access to data detailing the nature and extent of the follow-up these participants actually end up receiving, it is an explicit goal of NorCog that it aligns with national guidelines for dementia care [76]. For many participants, especially those with AUDs, the assessment might have triggered increased clinical attention, better care coordination, or referrals to other services. As mentioned in the introduction, individuals with AUDs are known to face risks associated with medication interactions, falls, and other somatic complications. It is possible that the initial high costs among those with AUDs reflect a period of deteriorating health or acute medical events leading up to the NorCog assessment. If such issues were identified and managed during the NorCog process, the subsequent need for acute and costly healthcare could have declined. Thus, while individuals with an AUD incurred higher healthcare costs than those without an AUD in the year preceding cognitive assessment, the significant result of our interaction analysis between AUD status and time shows that this difference diminished following assessment. This finding could have important clinical implications, as it implies that post-diagnostic care for cognitive impairment may simultaneously mitigate the healthcare burden associated with AUDs or its related sequelae.

Furthermore, alcohol consumption frequency was not associated with healthcare costs in sensitivity analyses. One explanation for this null result could stem from the common observation that older abstainers often have worse health than those who drink moderately more often [32,35,37]. We therefore speculate that this finding may reflect a so-called "sick quitter" effect, in which people with poor health are more likely to stop drinking precisely because of their health issues

[32,35,37]. Moreover, the very heaviest drinkers are less likely to enroll in scientific studies [70,97,98] and less likely to seek medical treatment in general [99,100]. Thus, it is likely that the very heaviest (or most frequent) drinkers were not included in our study, in which case our analyses would underestimate the true effects of alcohol consumption frequency on healthcare costs.

There are several strengths in this study. For one, our sample is large and includes diverse participants assessed across all Norwegian major health regions and by medical professionals. In addition, NorCog uses a robust selection of validated, standardized, and internationally recognized interviews and instruments. The high number of available variables permitted comprehensive adjustment for confounding variables. Furthermore, some of these instruments allowed us to examine variables not covered by previous research, such as the effect of the next of kin's relationship to the participant. This data was then linked with comprehensive and highly detailed objective assessments from national health registries, which permitted a robust exploration of the research questions raised in this study.

Some limitations should be addressed. Regarding representativity, people assessed for dementia in specialist healthcare in Norway are somewhat different from people assessed in primary healthcare, who are older, less educated, more cognitively impaired, have poorer ADL functioning, and more neuropsychiatric symptoms [101]. Thus, our findings may not generalize to individuals assessed solely in primary care or not referred for specialist evaluation. Furthermore, we did not have access to data related to home care nor the use of nursing homes, the latter of which is by far the largest cost driver among people with dementia [7,8]. We also excluded 830 participants (23.2%) due to missing or incompatible data, which reduces the overall representativeness of our sample. On the other hand, between-group analyses found no differences between the eligible and ineligible participants, indicating that the sample was an adequate representation of the registered participants. There are also limitations to the use of average unit costs in our calculations of specialist care costs, which may not capture individual resource use and thus introduce measurement error.

Furthermore, there are limitations to our independent variable related to AUDs. We operationalized the presence of AUDs by using ICD-10 diagnoses. These were derived by reviewing all ICD diagnoses registered in NorCog, KUHR, and NPR and extracting those with relevance to alcohol consumption. However, there may well exist other ICD diagnoses related to alcohol consumption which are unaccounted for in our study, due to the lack of NorCog participants afflicted with such conditions or due to the lack of precision in registered diagnoses. In addition, while some ICD diagnoses were per definition related to harmful or dependent alcohol consumption (e.g., F10: See Table 1), others were related to alcohol in more indirect or opaque ways (e.g., G31.2: See Table 1). On the one hand, this introduced heterogeneity in our AUD subsample that might have diluted our findings, on the other hand it increased statistical power due to a larger subgroup. Finally, there are important limitations to the subjective reports of alcohol consumption, which is a major reason why this variable was restricted for use in the sensitivity analyses. Specifically, people are frequently found to underreport their drinking [102–107], and frequency measures do not allow estimates of the precise amount of alcohol consumed on any given occasion.

## Conclusion

In conclusion, this study found approximately 50% higher healthcare costs among older adults being assessed for cognitive impairment with AUDs compared to those without AUDs. We also found that while individuals with an AUD incurred higher healthcare costs in the year preceding cognitive assessment, this difference significantly diminished following assessment and could imply that post-diagnostic care for cognitive impairment may simultaneously mitigate the healthcare burden associated with AUDs or its related sequelae. Our findings indicate that alcohol consumption is a potentially important and amenable determinant of healthcare use, knowledge which could also possibly contribute to curtailing the higher healthcare costs among older adults with AUDs specifically. We urge all healthcare providers to routinely ask their patients about alcohol consumption.

## Supporting information

**S1 Table. Unadjusted and adjusted linear regressions of the total costs one year before and after NorCog assessment, for both AUDs and alcohol consumption.**
(DOCX)

**S2 Table. Unadjusted and adjusted linear regressions of the total costs one year before and after NorCog assessment, for alcohol consumption only.**
(DOCX)

## Acknowledgments

The authors would like to thank all NorCog participants, colleagues working on NorCog, and the Norwegian National Centre for Ageing and Health. We would also like to thank the Norwegian Prescribed Drug Registry for conducting the linkage of the data from the NorCog, KUHR, and NPR databases. Disclaimer: The interpretation and reporting of the data from NorCog, NorPD, NPR, and KUHR are the sole responsibility of the authors, and no endorsement by NorCog, NorPD, NPR, or KUHR is intended nor should be inferred.

## Author contributions

**Conceptualization:** Ben Kamsvaag, Sverre Bergh, Anne-Sofie Helvik.

**Data curation:** Ben Kamsvaag.

**Formal analysis:** Ben Kamsvaag, Jūratė Šaltytė Benth.

**Funding acquisition:** Sverre Bergh, Anne-Sofie Helvik.

**Investigation:** Ben Kamsvaag, Sverre Bergh, Jūratė Šaltytė Benth, Anne-Sofie Helvik.

**Methodology:** Ben Kamsvaag, Sverre Bergh, Jūratė Šaltytė Benth, Anne-Sofie Helvik.

**Project administration:** Sverre Bergh, Anne-Sofie Helvik.

**Resources:** Sverre Bergh.

**Validation:** Jūratė Šaltytė Benth.

**Writing – original draft:** Ben Kamsvaag.

**Writing – review & editing:** Ben Kamsvaag, Sverre Bergh, Jūratė Šaltytė Benth, Richard C. Oude Voshaar, Kjerstin Tevik, Geir Selbaek, Anne-Sofie Helvik.

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
