## [Decision Letter · Decision Letter 0]

9 Dec 2025

Dear Dr.  Kamsvaag,

Thank you for submitting your manuscript to PLOS ONE. After careful consideration, we feel that it has merit but does not fully meet PLOS ONE’s publication criteria as it currently stands. Therefore, we invite you to submit a revised version of the manuscript that addresses the points raised during the review process.

We look forward to receiving your revised manuscript.

Kind regards,

Gihyun Yoon, MD

Academic Editor

PLOS One

Journal Requirements:

3. Thank you for uploading your study's underlying data set. Unfortunately, the repository you have noted in your Data Availability statement does not qualify as an acceptable data repository according to PLOS's standards.

Additional Editor Comments:

1. Abstract: Include the names of the primary outcome variables in the abstract section.

2. Abstract: Revise the sentence “Linear regression analysis was applied to assess the relationship between ICD-10 alcohol-related diagnoses and costs of (1) primary healthcare and (2) specialist health care.” to reduce confusion by emphasizing the primary outcome variable (healthcare costs). If the distinction between primary and specialist healthcare is not central, remove it from the abstract.

3. Figure 1 is difficult to interpret. Please revise it in the style of a CONSORT flowchart for clarity.

4. Page 19, line 302: Correct “pear year” to “per year.”

Reviewer's Responses to Questions

**Comments to the Author**

1. Is the manuscript technically sound, and do the data support the conclusions?

Reviewer #1: Yes

2. Has the statistical analysis been performed appropriately and rigorously?

Reviewer #1: Yes

3. Have the authors made all data underlying the findings in their manuscript fully available?

Reviewer #1: Yes

4. Is the manuscript presented in an intelligible fashion and written in standard English?

Reviewer #1: Yes

Reviewer #1: The authors have done a commendable job in presenting a complex analysis with a large data set in a way that is coherent, organized, scientifically sound, and yields understandable results. The discussion is balanced and has a clear take home message.

**Do you want your identity to be public for this peer review?** For information about this choice, including consent withdrawal, please see our Privacy Policy

Reviewer #1: No

---

## [Author Response · Author response to Decision Letter 1]

18 Dec 2025

Dear Editor and Reviewer. Thank you for your consideration and helpful feedback. We have made necessary adjustments according to the full list included in the e-mail sent to us on December 10th, and have organized our response according to each point of the list as follows:

1. PLOS ONE style requirements: The style requirements have been reviewed after making the necessary adjustments to the manuscript, and the manuscript should still be correct as far as we are aware. Thus, no changes were made.

2. Grant information conflict: As per the Editor’s request, we have now changed the name of the funding source to “Innlandet Hospital Trust” in the online submission system, to match the Financial Disclosure statement of the PDF build. Please note that this name was not automatically identified in the online submission system, so we had to enter it manually (only the Norwegian name, “Sykehuset Innlandet HF”, was listed). The grant number 150919 was correct in both instances (i.e., both Financial Disclosure and Funding Information), and was therefore left unchanged.

3. Data availability: Data cannot be shared publicly due to ethical and legal restrictions related to participant confidentiality, imposed by the Norwegian National Centre for Ageing and Health, the Norwegian Institute of Public Health, and the Data Protection Office of Innlandet Hospital Trust. Therefore, we have not attempted to upload our study’s underlying dataset; the previous data availability statement merely made a referral to the Norwegian Prescribed Drug Registry, where researchers who meet the criteria for access to confidential data may request access to the dataset. To make this distinction more clear, the data availability statement in the online submission system has been updated as follows:

“Data cannot be shared publicly due to ethical and legal restrictions related to participant confidentiality, imposed by the Norwegian National Centre for Ageing and Health, the Norwegian Institute of Public Health, and the Data Protection Office of Innlandet Hospital Trust. The dataset consists of sensitive, in-depth quantitative data regarding participants diagnosed with cognitive impairment and/or alcohol use disorders. Furthermore, participants consented to the use of their data solely within the scope of the present study, as approved by the South-Eastern Regional Committee for Medical Research Ethics (REK). Therefore, the dataset cannot be made publicly available. Researchers who meet the criteria for access to confidential data may request access to the dataset by contacting the institutional representatives at the Norwegian Prescribed Drug Registry, who were responsible for data linkage: lmr.data@fhi.no. For questions related to data privacy or ethical oversight, inquiries may be directed to Birgit Hovde at birgit.Hovde@sykehuset-innlandet.no (Data Protection Officer, Innlandet Hospital Trust).”

4. Previously published works: The reviewer comments included no recommendation to cite specific works.

5. Reference list: Only one new citation has been added to the manuscript, the 2025 CONSORT statement regarding the CONSORT flow chart standard (reference #94). The reference list has been reviewed and should be correct.

6. Addition Editor comments:

a. Abstract primary outcomes: Page 2, lines 29-30: As per the Editor’s request, the primary outcome variable has now been named explicitly by revising the end of the sentence to “…the primary outcome variable: healthcare costs.”

b. Abstract revision: Page 2, lines 29-30: As per the Editor’s suggestion, we have removed the distinction between the primary and specialist healthcare costs from the abstract, as it is not central to our study.

c. Figure 1: Figure 1 has been revised according to CONSORT flow chart standards, as per the Editor’s request. The flow chart layout had to be adapted to match our research design, as our study is not a randomized trial. Lines 140-143 (page 7) were also revised to address this change.

d. Page 19 line 304: “Pear year” has been corrected to “per year”.

---

## [Editor Report · Decision Letter 1]

22 Dec 2025

Alcohol use disorders are associated with higher healthcare expenditure among older adults with suspected cognitive impairment: A registry-based cross-sectional study

PONE-D-25-59078R1

Dear Dr. Kamsvaag,

We’re pleased to inform you that your manuscript has been judged scientifically suitable for publication and will be formally accepted for publication once it meets all outstanding technical requirements.

Kind regards,

Gihyun Yoon, MD

Academic Editor

PLOS One
---

## [Editor Report · Acceptance letter]

PONE-D-25-59078R1

PLOS One

Dear Dr. Kamsvaag,

I'm pleased to inform you that your manuscript has been deemed suitable for publication in PLOS One. Congratulations! Your manuscript is now being handed over to our production team.

Kind regards,

on behalf of

Dr. Gihyun Yoon

Academic Editor

PLOS One